# Practical iridium-catalyzed direct α-arylation of N-heteroarenes with (hetero)arylboronic acids by H$_2$O-mediated H$_2$ evolution

Liang Cao[1], He Zhao[1], Rongqing Guan[1], Huanfeng Jiang ⬤ [1], Pierre. H. Dixneuf[2] & Min Zhang ⬤ [1✉]

Despite the widespread applications of 2-(hetero)aryl N-heteroarenes in numerous fields of science and technology, universal access to such compounds is hampered due to the lack of a general method for their synthesis. Herein, by a H$_2$O-mediated H$_2$-evolution cross-coupling strategy, we report an iridium(III)-catalyzed facile method to direct α-arylation of N-heteroarenes with both aryl and heteroaryl boronic acids, proceeding with broad substrate scope and excellent functional compatibility, oxidant and reductant-free conditions, operational simplicity, easy scalability, and no need for prefunctionalization of N-heteroarenes. This method is applicable for structural modification of biomedical molecules, and offers a practical route for direct access to 2-(hetero)aryl N-heteroarenes, a class of potential cyclometalated C^N ligands and N^N bidentate ligands that are difficult to prepare with the existing α-C-H arylation methods, thus filling an important gap in the capabilities of synthetic organic chemistry.

[1] Key Lab of Functional Molecular Engineering of Guangdong Province, School of Chemistry and Chemical Engineering, South China University of Technology, Guangzhou, China. [2] University of Rennes, ISCR, Rennes, France. ✉email: minzhang@scut.edu.cn

2-(Hetero)aryl N-heteroarenes represent a class of important compounds in numerous fields of science and technology, as they are extensively applied for the development of bioactive molecules, drugs, functional materials, ligands, and chemosensors[1–3]. For instance, N-Heteroarenes 1-3 illustrated exhibit diverse interesting bioactivities (Fig. 1)[4–6]. Selexipag (uptravi) 4 as a top-selling drug is used for the treatment of cardiovascular diseases[7,8]. 2-Pyridyl N-heteroarenes 5 possess unique binding capability towards various metals, which make them highly useful bidentate ligands in catalysis and organometallic chemistry[9–11]. In addition, 2-aryl N-heteroarenes also play a key role in photochemistry and functional materials[12–17], as they can serve as C^N ligands to generate cyclometalated complexes with diverse photophysical properties (Fig. 1 example 6).

Due to the widespread applications, the introduction of (hetero)aryl groups to the α-site of N-heteroarenes is of significant importance, as it enables key step to access various desired 2-(hetero)aryl N-heteroarenes. Conventionally, such compounds are synthesized by Pd-catalyzed Suzuki cross-coupling of 2-halogenated N-heteroarenes with arylboronic acids[18]. However, the halo substrates used are often hard to prepare due to the difficulties in the control of the chemo- and regioselectivity during the halogenation processes. Later, the C–C cross-coupling at C2-position of quinolines or related N-heterocycles was achieved with ArZnEt and Ni(0) catalyst[19,20], or with ArMgX by using Fe(III)[21] or Co(II) catalyst[22,23], or preferentially with aryl bromides in the presence of Rh(I) catalyst but at 175–190 °C[24,25] (Fig. 2a). Nevertheless, the need for high reaction temperatures or stringent protecting operations toward air and moisture-sensitive organometallic agents limit the practicality of these synthetic protocols. In recent years, Minisci-type radical coupling has also been nicely employed to arylate the α-C–H bond of N-heteroarenes (Fig. 2b)[26–31], but the related transformations generally produce several regioisomers, and consume excess of less environmentally benign oxidants ($K_2S_2O_8$ and Selectfluor). The substrates containing oxidant-sensitive groups (e.g., –$NR_2$ and –SR) do not allow to afford the desired products. Moreover, all the above-described α-C-H arylation protocols[19–31] are incompatible with heteroaryl bromides, metallic agents, and carboxylates, thus the preparation of 2-heteroaryl N-heteroarenes including N^N bidentate ligands is restricted. In this context, there is a high demand for strategies enabling the direct and efficient introduction of both aryl and heteroaryl groups into the α-site of N-heteroarenes, preferably with readily available and stable feedstocks [32,33].

Inspired by our recent discovery of hydrogen-transfer-mediated α-functionalization of 1,8-Naphthyridines with

tetrahydroquinolines under iridium catalysis (Fig. 3a)[34], we were motivated to test a reductive α-arylation of non-activated quinoline A1 with p-tolylboronic acid B1. However, with the same iridium(III) catalyst system, the reaction of A1 and B1 in t-amyl alcohol employing different reductants (such as i-PrOH[35–37], $NH_3BH_3$[38], Hantzsch esters[39,40], $HCO_2H$[41], and $HCO_2Na$[42,43]) all failed to afford the desired 2-aryl tetrahydroquinoline C1′ (Fig. 3b). Interestingly, the absence of reductant resulted in the production of 2-(p-tolyl)quinoline C1 in 22% yield at 110 °C.

Here, we wish to report a practical iridium-catalyzed direct α-arylation of N-heteroarenes with both aryl and heteroarylboronic acids by a $H_2O$-mediated hydrogen-evolution cross-coupling strategy (Fig. 3c), which offers a practical platform for direct structural modification of pyridine-containing molecules including drugs and functional materials, and facile preparation of N-heteroarene bidentate ligands as well.

## Results

**Investigation of reaction conditions**. Initially, we wished to screen an efficient reaction system and the coupling of substrates A1 and B1 was chosen as a model system to evaluate different parameters (Table 1). At first, the reaction in t-amyl alcohol was performed at 110 °C for 24 h by testing different catalyst precursors (Ir(III), Ir(I), Ru(0), and Pd(II)). $[Cp*IrCl_2]_2$ exhibited the best performance to afford product C1 in 22% yield (entries 1–4). So, $[Cp*IrCl_2]_2$ was utilized to further evaluate a series of additives (entries 5–8), the results showed that the bases had a detrimental effect on the reaction (entries 5 and 6), whereas amino acids, such as glycine and L-proline, significantly improved the product yields, and the use of 20 mol% L-proline showed to be the best choice (entries 7 and 8). Then, we tested different solvents, we noticed that the reaction performed in dry 1,4-dioxane failed to produce any product C1 (entry 9), whereas the use of aqueous solution significantly increased the product yield to 60% (entry 10), which clearly implies that the presence of $H_2O$ plays a decisive role on the product formation. Interestingly, the mixed solution of $H_2O$ and 1,4-dioxane (v/v = 10/1) further improved the yield to 72% (entry 11). However, change of volume ratios was unable to further increase the product yield (entry 12). In comparison, $H_2O$ in combination with other solvents in a volume ratio of 10: 1 showed to be inferior to the mixed solution of $H_2O$ and 1,4-dioxane (entries 13–15). Decrease or increase of the reaction temperature also failed to improve the reaction efficiency (entry 16). The blank experiments indicated that only the presence of both $[Cp*IrCl_2]_2$ and L-proline can constitute an efficient catalyst system (entries 17 and18). Finally, the application of other iridium catalysts showed that they were inferior to $[Cp*IrCl_2]_2$ (entry 19). Hence, the optimal conditions are as shown in entry 11 when the reaction is performed in mixed $H_2O$ and 1,4-dioxane solution (v/v = 10/1) at 110 °C for 24 h in the presence of 1 mol% of $[Cp*IrCl_2]_2$ and 20 mol% of L-proline.

**Substrate scope**. With the optimal reaction conditions in hand, we then examined the generality of the synthetic method. First, quinoline A1 in combination with a wide array of arylboronic acids B (see Supplementary Fig. 1 in Supplementary Information (SI) for structural information) were examined. As illustrated in Fig. 4, all the reactions proceeded smoothly and furnished the desired products in good to excellent isolated yields (C2–C28), these products have the potential to serve as C^N ligands and generate cyclometalates[16,17]. Interestingly, a variety of functionalities (i.e., alkyl, –OMe, –SMe, –F, –Cl, –Br, –$SiMe_3$, –COMe, –$CO_2Et$, –$CF_3$, –$NO_2$, acetal, –OPh, and –$NPh_2$) on the aryl rings of boronic acids were well tolerated, and the retention of these functional groups offers the potential for molecular complexity

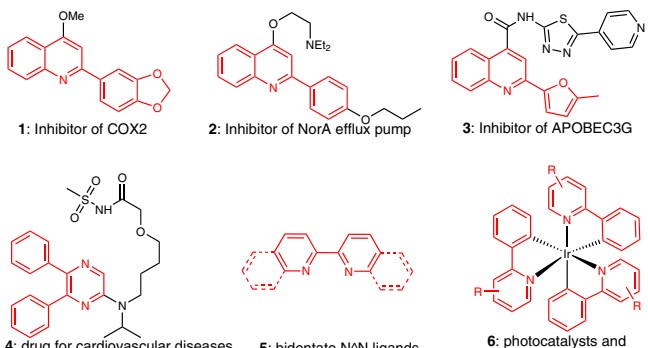

**Fig. 1 Selected examples containing useful 2-(hetero)aryl N-heteroarenes.** Structurally related pharmaceuticals, ligands, and photocatalyst.

1: Inhibitor of COX2
2: Inhibitor of NorA efflux pump
3: Inhibitor of APOBEC3G
4: drug for cardiovascular diseases
5: bidentate N^N ligands
6: photocatalysts and optoelectronic materials

**Fig. 2 Previous methods for access to 2-aryl N-heteroarenes. a** Transition metal-mediated C-H arylation of N-heteroarenes. **b** Minisci-type radical arylation of N-heteroarenes.

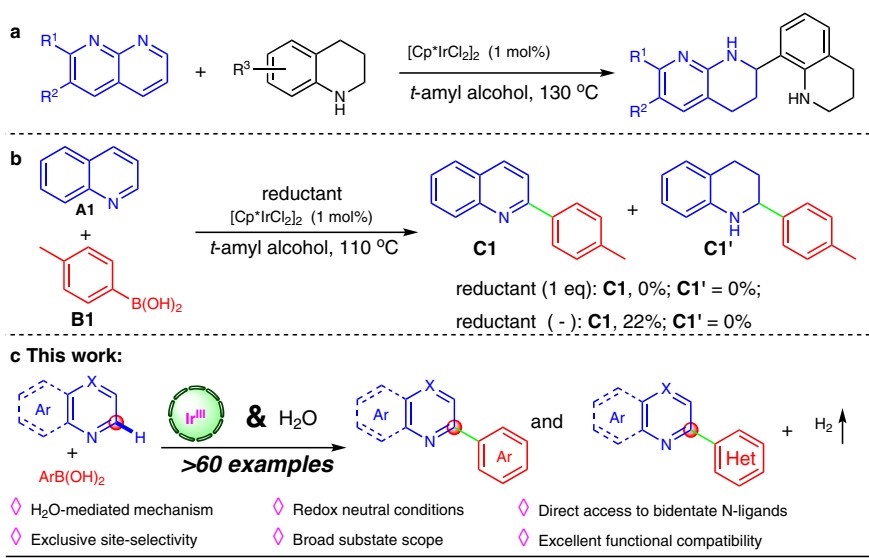

**Fig. 3 Observation on direct α-arylation of quinoline. a** Hydrogen-transfer-mediated α-functionalization of 1,8-naphthyridines with tetrahydroquinolines under iridium catalysis. **b** Attempts on iridium-catalyzed α-arylation of quinoline with *p*-tolyboronic acid. **c** General iridium-catalyzed direct α-(hetero) arylation of N-heteroarenes.

via further chemical transformations. In general, arylboronic acids bearing electron-donating groups (**C4**–**C6**, **C8**–**C9**, and **C20**–**C22**) afforded the products in higher yields than those of arylboronic acids with strong electron-withdrawing groups (**C15**–**C19**), implying that the reaction involves a nucleophilic coupling step. Besides, *ortho*-substituted arylboronic acids resulted in relatively lower yields (**C3**, **C7**, and **C10**), showing that the steric hindrance has a certain influence on the reaction. In addition to arylboronic acids, heteroaryl boronic acids such as indolyl, pyridyl, furanyl, and thiophenyl ones (**B24**–**B28**) were also amenable to the transformation, affording the desired 2-heteroaryl N-heteroarenes in moderate yields (**C24**–**C28**).

Then, we screened the reaction with a variety of N-heteroarenes (**A2**-**A22**, see Supplementary Fig. 1 for their structures) employing *p*-tolyboronic acid **B1**. First, a variety of quinolines with different substitution patterns (**A2**–**A18**) were tested. As illustrated in Fig. 5, all the substrates underwent smooth cross-coupling to generate the desired products in moderate to excellent yields upon isolation (**C29**–**C45**). A series of functional groups on quinolyl skeleton (i.e., –Me, –OMe, –F, –Cl, –Br, –I, –CO₂Me, –NO₂) were also well tolerated, and N-heteroarenes containing electro-withdrawing groups gave relatively higher yields (**C34**-**C37**, **C44** and **C45**) than those of electron-rich ones (**C33** and **C43**), which is rationalized as the

electron-deficient quinolines are beneficial to nucleophilic coupling with arylboronic acids. Except for quinoline derivatives, other types of N-heteroarenes such as quinoxaline, quinazoline, 1,5-naphthyridine, 1,8-naphthyridine, imidazo[1,2-*a*]pyrazine, 7,8-benzoquinoline, phenanthridine, and thieno[3,2-*b*] pyridine (**A19**-**A26**) were also compatible coupling partners to react with *p*-tolyboronic acid **B1**, delivering the desired cross-coupling products in reasonable yields (**C46**–**C53**). Noteworthy, reactants **A19**-**A22** have two reactive α-sites, but the reaction only generated *mono*-arylated products even in the presence of excess boronic acids, showing that the reaction merits unique chemoselectivity. In addition, the more challenging pyrimidine and pyrazine can also give the corresponding products **C54** and **C55** by prolonging the reaction time. Interestingly, by introducing 20 mol% of CF₃COOH as the activating agent, the α-arylation of pyridine derivatives was also successful, albeit the product yields were somewhat low (**C56**-**C58**). As shown in Figs. 4, 5, the demonstrated examples indicate that the synthetic protocol showed broad substrate scope and excellent functional group compatibility, regardless of oxidant and acid-sensitive ones (**C9** and **C23**).

The preparation of N-bidentate ligands with the existing C–H arylation protocols still remains an unresolved goal due to the difficulties in the preparation of 2-heteroaryl organometallic

**Table 1 Optimization of reaction conditions[a].**

| Entry | Catalyst | Additive | Solvent | C1 (%)[b] |
|---|---|---|---|---|
| 1 | [Cp*IrCl$_2$]$_2$ | – | t-AmOH | 22 |
| 2 | [IrCl(cod)]$_2$ | – | t-AmOH | <5 |
| 3 | Ru$_3$(CO)$_{12}$ | – | t-AmOH | 0 |
| 4 | Pd(OAc)$_2$ | – | t-AmOH | 0 |
| 5 | [Cp*IrCl$_2$]$_2$ | K$_3$PO$_4$ | t-AmOH | Trace |
| 6 | [Cp*IrCl$_2$]$_2$ | Cs$_2$CO$_3$ | t-AmOH | Trace |
| 7 | [Cp*IrCl$_2$]$_2$ | Glycine | t-AmOH | 35 |
| 8 | [Cp*IrCl$_2$]$_2$ | L-proline | t-AmOH | (37, 40, 35)[c] |
| 9 | [Cp*IrCl$_2$]$_2$ | L-proline | dry 1,4-dioxane | – |
| 10 | [Cp*IrCl$_2$]$_2$ | L-proline | H$_2$O | 60 |
| 11 | [Cp*IrCl$_2$]$_2$ | L-proline | H$_2$O/1,4-dioxane | 72[d] |
| 12 | [Cp*IrCl$_2$]$_2$ | L-proline | H$_2$O/1,4-dioxane | (66, 70)[e] |
| 13 | [Cp*IrCl$_2$]$_2$ | L-proline | H$_2$O/t-AmOH | 40 |
| 14 | [Cp*IrCl$_2$]$_2$ | L-proline | H$_2$O/DMSO | 35 |
| 15 | [Cp*IrCl$_2$]$_2$ | L-proline | H$_2$O/DMF | 30 |
| 16 | [Cp*IrCl$_2$]$_2$ | L-proline | H$_2$O/1,4-dioxane | (65, 72)[f] |
| 17 | – | L-proline | H$_2$O/1,4-dioxane | 0 |
| 18 | [Cp*IrCl$_2$]$_2$ | – | H$_2$O/1,4-dioxane | 48 |
| 19 | Ir complexes | L-Proline | H$_2$O/1,4-dioxane | (<5, <5, 32)[g] |

Cp* 1,2,3,4,5-pentamethylcyclopentadiene, cod 1,5-cyclooctadiene, DMSO dimethyl sulfoxide, DMF N,N-dimethylformamide
[a] Unless otherwise stated, the reaction in t-amyl alcohol (1.5 mL) was performed with **A1** (0.3 mmol), **B1** (0.36 mmol), catalyst (1 mol%), additive (20 mol%) at 110 °C for 24 h under N$_2$
[b] Isolated yield
[c] Yields are with respect to use of 10 mol%, 20 mol%, and 40 mol% L-proline, respectively
[d] Mixed H$_2$O and 1,4-dioxance solution in a volume ratio of 10:1
[e] Yields are with respect to used mixed H$_2$O and 1,4-dioxane solution in volume ratios of 9:1 and 11:1, respectively
[f] Yields are with respect to the temperatures at 100 °C and 120 °C, respectively.
[g] Yields are with respect to use of catalyst [IrCl(cod)]$_2$, [Ir(OMe)(1,5-cod)]$_2$, IrCl$_3$·3H$_2$O, respectively

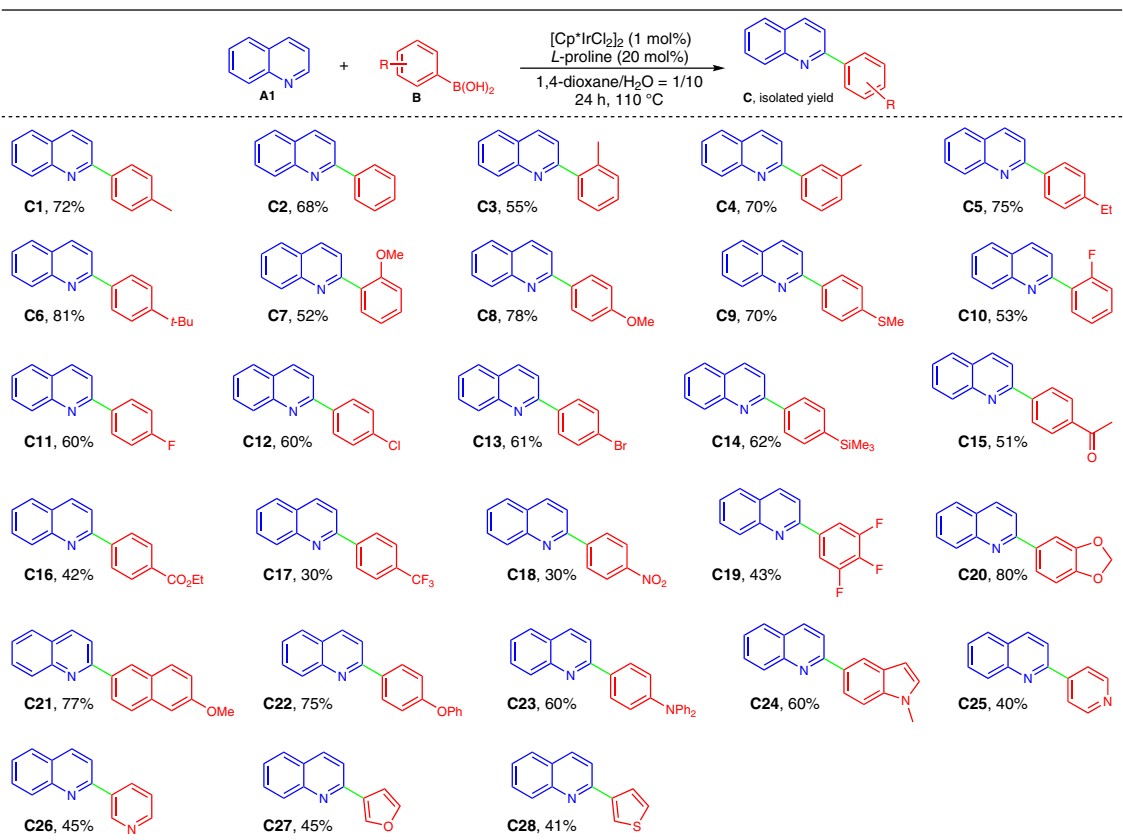

**Fig. 4 Synthesis of 2-aryl quinolines by variation of arylboronic acids.** Reactions were conducted on a 0.3 mmol scale under the standard conditions. The isolated yields are reported.

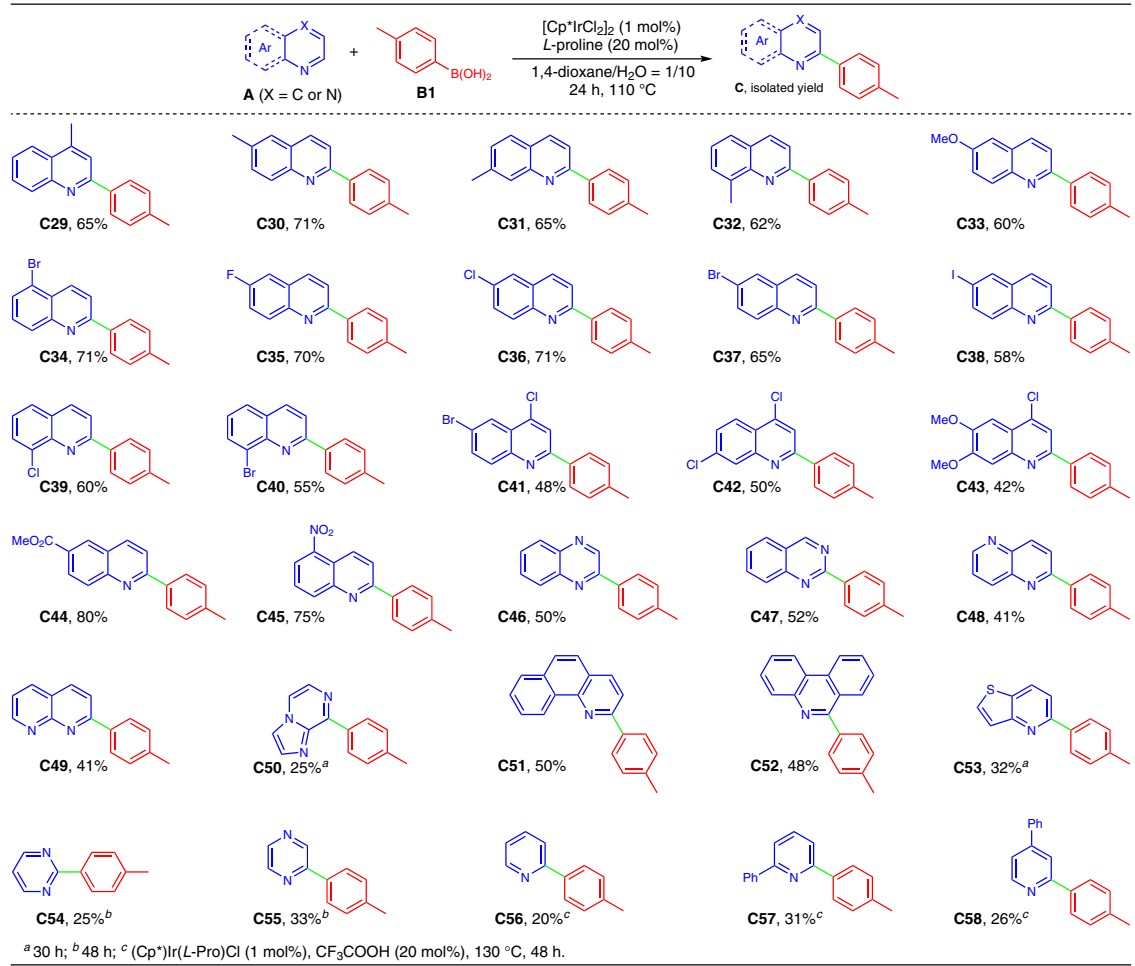

**Fig. 5 Synthesis of 2-*p*-tolyl products by variation of N-heteroarenes.** Reactions were conducted on a 0.3 mmol scale under the standard conditions. The isolated yields are reported.

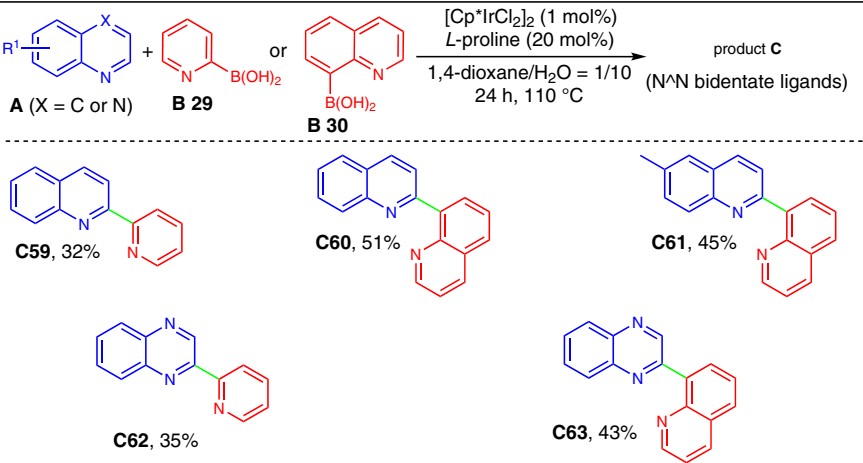

**Fig. 6 Direct access to different N-bidentate ligands by α-heteroarylation of N-heteroarenes.** Reactions were conducted on a 0.3 mmol scale under the standard conditions. The isolated yields are reported.

reagents and in situ formation 2-heteroaryl radicals[19–31]. Herein, we successfully addressed such an issue by utilizing our synthetic method. As shown in Fig. 6, representative pyridin-2-yl and quinolin-8-yl boronic acids (**B29** and **B30**) were employed to react with quinoline **A1** and quinoxaline **A19**, respectively. All the reactions smoothly afforded the desired cross-coupling products in moderate yields. Interestingly, these obtained N^N bidentate ligands (**C59**–**C63**) and the commercially available 2,2′-bipyridine as well as 1,10-phenanthroline all did not undergo further α-arylation even in the presence of excess arylboronic acids, presumably because they can coordinate to the Ir(III) catalyst, and hamper the participation of Ir(III) in activation of

**Fig. 7 Control experiments. a** Verification experiment for intermediate analysis. **b** Kinetic isotope effect experiment. **c** Detection of B(OH)$_3$. **d** Detection of H$_2$. **e** Synthesis of the active catalyst Cp*Ir(L-Pro)Cl. **f** Verification experiments for the active catalyst Cp*Ir(L-Pro)Cl.

these bis-nitrogen heteroarenes. Thus, the present work offers a practical platform for the direct and selective preparation of valuable N-bidentate ligands[9–11].

**Mechanistic investigations.** To gain mechanistic insights into the α-C–H arylation reaction, several control experiments were carried out (Fig. 7). First, the model reaction does not occur at all in the absence of Ir(III) catalyst (Table 1, entry 17), and both 1,2,3,4-tetrahydroquinoline (**A1-a**) and dihydroquinolines (**A1-b** and **A1-c**) were unable to couple with p-tolyboronic acid (**B1**) to yield product **C1** (Fig. 7a) under the standard conditions, showing that the reaction involving tetrahydroquinoline and dihydroquinoline as the intermediates is not likely, as it was the case for reductive cross-coupling of N-heterocycles in t-amyl alcohols[34], and the catalyst plays a crucial role in initiating the reaction. Upon a concurrent competition experiment of p-tolyboronic acid **B1** with quinoline **A1** and its α-deuterated counterpart **A1-d** (Supplementary Fig. 2), [1]H-NMR analysis showed a kinetic isotope effect (KIE) value of 1.4 (Supplementary Fig. 3), indicating that the cleavage of α-C-H bond of quinoline **A1** is not the rate-determining step in the reaction (Fig. 7b). Noteworthy, after completion of the reaction, B(OH)$_3$ and H$_2$ by-products[44–46] were detected by means of [11]B-NMR and GC, respectively (Figs. 7c and 7d, see Supplementary Figs. 4 and 5, Supplementary Table 1). To further understand the role of L-proline in the reaction, we prepared complex Cp*Ir(L-Pro)Cl from [Cp*IrCl$_2$]$_2$ and L-proline (Fig. 7e). The application of Cp*Ir(L-Pro)Cl in the model reaction resulted in product **C1** in 75% yield, whereas the combination of this complex with additional L-proline failed to improve the product yields (Fig. 7f). In comparison, the use of [Cp*IrCl$_2$]$_2$ catalyst without L-proline only gave a 48% product yield (Table 1, entry 18). These experiments indicate that Cp*Ir

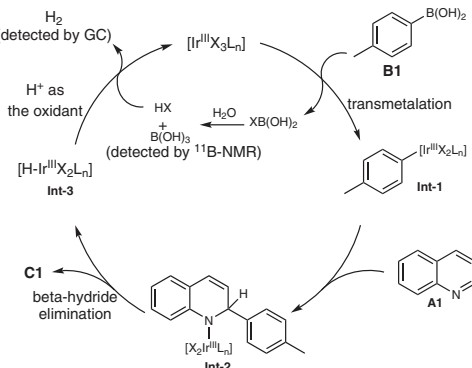

**Fig. 8 Plausible reaction mechanism.** Iridium-catalyzed H$_2$O-mediated H$_2$ evolution α-arylation of quinoline with p-tolyboronic acid.

(L-Pro)Cl is the reaction catalyst, and L-proline serves as a ligand to form the iridium catalyst.

Although the mechanistic details have not been fully elucidated, a plausible reaction pathway for the model reaction is depicted in Fig. 8 based on the above-described findings. Initially, the L-proline serves as a ligand[47–49] of Ir(III) metal species (Fig. 7e) to form the complex [Ir$^{III}$X$_3$L$_n$]. The transmetalation[19,20] between p-tolyboronic acid **B1** and [Ir$^{III}$X$_3$L$_n$] forms aryl-Ir complex **Int-1** with the elimination of XB(OH)$_2$. The metathesis of XB(OH)$_2$ and H$_2$O produces HX and B(OH)$_3$ (detected by [11]B-NMR, Supplementary Fig. 4). Then, quinoline **A1** undergoes carbon-Ir bond insertion of complex **Int-1** into its imino motif (**Int-2**), and the subsequent β-hydride elimination from **Int-2** gives rise to the desired product **C1** along with the generation of metal hydride species [H-Ir$^{III}$X$_2$L$_n$] (**Int-**

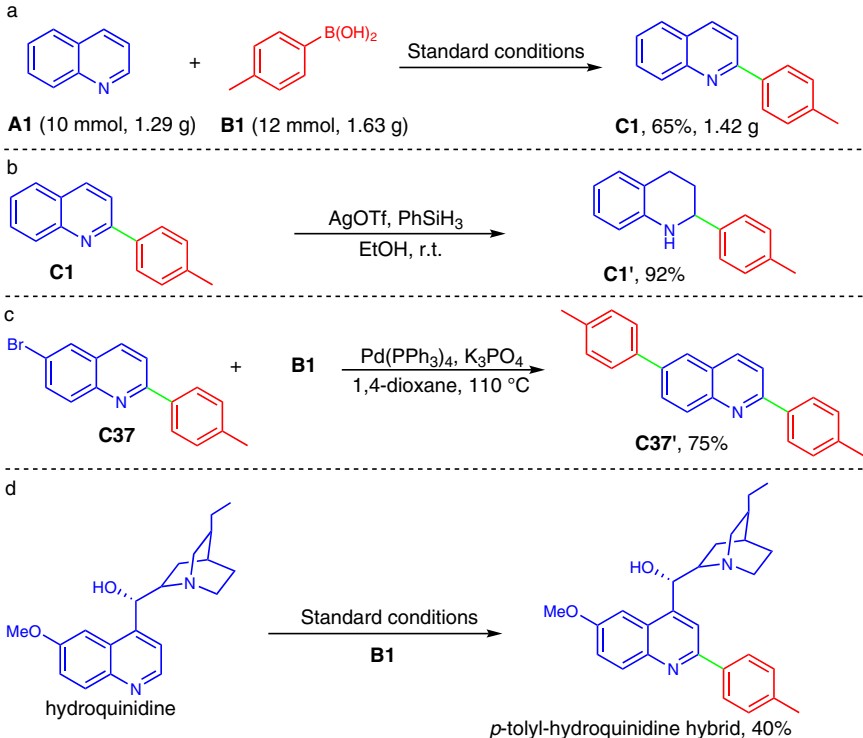

**Fig. 9 Synthetic utility of the developed chemistry. a** Gram-scale synthesis. **b** Reduction. **c** Cross-coupling. **d** Structural modification of biomedical molecule.

**3**). Finally, the interaction of the hydride in [H-Ir$^{III}$X$_2$L$_n$] with HX (as an oxidant) would regenerate the iridium(III) catalyst and liberate H$_2$ gas (detected by gas chromatography, Supplementary Fig. 5). In the whole catalytic cycle, H$_2$O-mediated H$_2$ evolution plays a crucial role in facilitating the transmetalation process and regenerating the catalyst. The profitable role of the proline is likely coordinated through its carboxylate to Ir(III) as a X ligand as in copper(I) catalyst [44-47] and as a L ligand via its R$^2$NH group (Fig. 7e).

**Application**. Finally, we were interested in demonstrating the synthetic utility of the developed chemistry. As shown in Fig. 9, gram-scale synthesis of 2-arylquinoline **C1** (1.42 g) was achieved by scaling up substrates **A1** and **B1** to 10 mmol and 12 mmol, respectively (Fig. 9a), and the reaction still afforded a desirable isolated yield (65%). Meanwhile, the transfer hydrogenation of compound **C1** produced a synthetically useful tetrahydroquinoline[50] **C1'** in excellent yield (Fig. 9b). Brominated compound **C37** underwent smooth Suzuki cross-coupling to afford arylated product **C37'** in 75% yield (Fig. 9c). Moreover, the reaction is also applicable for structural functionalization of biomedical molecule such as hydroquinidine, delivering the desired *p*-tolyl-hydroquinidine hybrid in 40% yields (Fig. 9d).

**Discussion**

In conclusion, by a H$_2$O-mediated H$_2$-evolution cross-coupling strategy, we have developed an iridium(III)-catalyzed direct α-arylation of non-activated N-heteroarenes with both aryl and heteroaryl boronic acids. This chemical avenue to 2-(hetero)aryl N-heteroarenes proceeds with broad substrate scope and excellent functional compatibility under redox neutral conditions, is operationally simple, scalable, and applicable for structural modification of biomedical molecules, enables direct access to useful bidentate N-ligands that are inaccessible or difficult to prepare with the existing α-C–H arylation protocols, and does not need for prefunctionalization of N-heteroarenes, which fills an important gap in the capabilities of synthetic organic chemistry. This catalytic reaction is anticipated to be applied in numerous fields of science and technology due to the promising potentials of 2-(hetero)aryl N-heteroarenes. Moreover, the strategy employed should be useful in the functionalization of other unsaturated hydrocarbons and further design of other reactions.

**Methods**

**Typical procedure I for the synthesis of α-arylation of N-heteroarenes**. Under N$_2$ atmosphere, [Cp*IrCl$_2$]$_2$ (1 mol%), *L*-proline (20 mol%), N-heteroarenes **A** (0.3 mmol), arylboronic acids **B** (0.36 mmol) and H$_2$O/1,4-dioxane (10/1, 1.5 mL) were introduced in a Schlenk tube (50 mL), successively. Then, the Schlenk tube was closed and the resulting mixture was stirred at 110 °C (oil bath temperature) for 24 h. After cooling down to room temperature, quenched with water, extracted with ethyl acetate (3 × 5 mL), and dried over anhydrous Na$_2$SO$_4$. The reaction mixture was concentrated by removing the solvent under vacuum, and the residue was purified by preparative TLC on silica, eluting with petroleum ether (60–90 °C) and ethyl acetate to give the desired product **C**.

**Data availability**

The authors declare that all relevant data supporting the findings of this study are available within the paper and its supplementary information files.

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

## Acknowledgements

The authors thank the National Key Research and Development Program of China (2016YFA0602900), National Natural Science Foundation of China (21971071), Natural Science Foundation of Guangdong Province (2021A1515010155), and the Fundamental Research Funds for the Central Universities (2020ZYGXZR075) for financial support.

## Author contributions

M.Z. and L.C. conceived the idea, analyzed the data, M.Z. wrote the manuscript and directed the project. H.Z. carried out the hydrogen test experiment. R.-Q.G. synthesized part of the raw material. H.-F.J. and P.H.D. revised the manuscript and discussed the mechanistic details. All the authors have read the manuscript and agree with its content.

## Competing interests

The authors declare no competing interests.
