## [Peer Review File · Nature Communications]

Reviewers' Comments:

Reviewer #1:

Remarks to the Author:

This submission by Zhang and co-workers describes the use of $[\text{CpIrCl}_2]_2$ as catalyst for direct arylation of N-heteroarenes with arylboronic acids. 1 mol% loadings of Ir catalyst in cooperation with 20 mol% L-proline as co-catalyst was shown to couple a diverse range of functionalized arylboronic acids with quinoline.

While the catalysis may require fewer additives compared to the literature precedents, the overall yields are only as good or even poorer than existing reports, including metal free processes using quinoline N-oxides, see: *Org. Lett.* 2015, 17, 3134–3137.

I am impressed by the excellent mechanistic work in this paper and the supporting information is very detailed and well presented. It is a bit disappointing that the role of L-proline still remains a mystery.

The exploration of the Ir catalyst should be expanded. $[\text{IrCl}(\text{COD})]_2$ gives low conversion in the absence of L-proline, but how does it compare with it? Also, how effective is Cp^*IrCl_2 without any additive? These reactions should be added to Table 1.

The work is well done, but the results obtained are not a significant improvement on related reactions published. There are some areas where the language and grammar need improvement, so I recommend some careful revision of the English usage prior to submission in a more specialized journal on organic synthesis or catalysis.

Reviewer #2:

Remarks to the Author:

This is a very nice chemistry to carry out the oxidative coupling between heterocycles and arylboronic acids via Ir catalysis. Especially a new pathway was observed by evolving the dihydrogen. The chemistry showed its potential application. Several points must be addressed before publications.

1. The substrate scope is relatively narrow although the potential applications have been demonstrated. How about other heterocycles, in particular pyridine derivatives?
2. In the catalytic cycle, H^+ seems to play a role as an oxidant to make the oxidation take place to generate H_2 . Please give the clear demonstration in the catalytic cycle.
3. Finally, the reference of the first example to demonstrate the oxidative addition between heterocycles and arylboronic acids might be better cited in the text (*Angew. Chem. Int. Ed.* 2008, 47, 1473-1476.)

Reviewer #3:

Remarks to the Author:

The manuscript by Zhang and co-workers describes a simple method for the alpha-arylation of nitrogen heterocycles. The compounds available by this method are very interesting from a pharmaceutical point of view and some of them are not readily available by alternative methodologies. Furthermore, the manuscript is well written. I recommend its acceptance with some corrections and clarifications.

- In Scheme 1, the use of a reducing agent should be added to the arrows.

- Have the authors tried the use of pyridine derivatives as substrates? This would lend a higher generality to their method. They have showcased the use of other single-ring heterocycles such as pyrimidines and pyrazines.

- Line 163. Please explain the KIE acronym.

REVIEWER COMMENTS

Reviewer #1 (Remarks to the Author):

This submission by Zhang and co-workers describes the use of [CpIrCl₂]₂ as catalyst for direct arylation of N-heteroarenes with arylboronic acids. 1 mol% loadings of Ir catalyst in cooperation with 20 mol% L-proline as co-catalyst was shown to couple a diverse range of functionalized arylboronic acids with quinoline. While the catalysis may require fewer additives compared to the literature precedents, the overall yields are only as good or even poorer than existing reports, including metal free processes using quinoline N-oxides, see: *Org. Lett.* 2015, 17, 3134–3137. I am impressed by the excellent mechanistic work in this paper and the supporting information is very detailed and well presented. It is a bit disappointing that the role of L-proline still remains a mystery.

The exploration of the Ir catalyst should be expanded. [IrCl(COD)]₂ gives low conversion in the absence of L-proline, but how does it compare with it? Also, how effective is Cp*IrCl₂ without any additive? These reactions should be added to Table 1.

The work is well done, but the results obtained are not a significant improvement on related reactions published. There are some areas where the language and grammar need improvement, so I recommend some careful revision of the English usage prior to submission in a more specialized journal on organic synthesis or catalysis.

Reviewer #2 (Remarks to the Author):

This is a very nice chemistry to carry out the oxidative coupling between heterocycles and arylboronic acids via Ir catalysis. Especially a new pathway was observed by evolving the dihydrogen. The chemistry showed its potential application. Several points must be addressed before publications.

1. The substrate scope is relatively narrow although the potential applications have been demonstrated. How about other heterocycles, in particular pyridine derivatives?
2. In the catalytic cycle, H⁺ seems to play a role as an oxidant to make the oxidation take place to generate H₂. Please give the clear demonstration in the catalytic cycle.
3. Finally, the reference of the first example to demonstrate the oxidative addition between heterocycles and arylboronic acids might be better cited in the text (*Angew. Chem. Int. Ed.* 2008, 47, 1473-1476.).

Reviewer #3 (Remarks to the Author):

The manuscript by Zhang and co-workers describes a simple method for the alpha-arylation of nitrogen heterocycles. The compounds available by this method are very interesting from a pharmaceutical point of view and some of them are not readily available by alternative methodologies. Furthermore, the manuscript is well written. I recommend its acceptance with some corrections and clarifications.

- In Scheme 1, the use of a reducing agent should be added to the arrows.
- Have the authors tried the use of pyridine derivatives as substrates? This would lend a higher generality to they method. They have showcased the use of other single-ring heterocycles such as pyrimidines and pyrazines.
- Line 163. Please explain the KIE acronym.

Response to Reviewers

Reviewer #1 (Remarks to the Author):

1. This submission by Zhang and co-workers describes the use of $[\text{CpIrCl}_2]_2$ as catalyst for direct arylation of N-heteroarenes with arylboronic acids. 1 mol% loadings of Ir catalyst in cooperation with 20 mol% L-proline as co-catalyst was shown to couple a diverse range of functionalized arylboronic acids with quinoline. While the catalysis may require fewer additives compared to the literature precedents, the overall yields are only as good or even poorer than existing reports, including metal free processes using quinoline N-oxides, see: *Org. Lett.* 2015, 17, 3134–3137. I am impressed by the excellent mechanistic work in this paper and the supporting information is very detailed and well presented. It is a bit disappointing that the role of L-proline still remains a mystery.

Response: Thank you very much for the positive comments and constructive questions. We appreciate the nice early work contributed by the authors (*Org. Lett.* 2015, 17, 3134–3137), which presents a metal-free cross-coupling of quinoline N-oxides with boronic acids. This reference is cited as ref 33. In comparison with this contribution and other related α -arylation shown in Scheme 1 of the manuscript, we wish to point out that the yields of our work are synthetically useful due to the far broader substrate scope and direct arylation of N-heteroarenes without the needs for the pre-preparation of specific coupling agents (e.g., quinoline N-oxides, organozinc and Grignard agents) and the use of less-environmentally benign oxidants and elevated reaction temperatures.

Regarding the role of L-proline in the reaction, we conducted the related control experiments. We have prepared $\text{Cp}^*\text{Ir}(L\text{-Pro})\text{Cl}$ complex from $[\text{Cp}^*\text{IrCl}_2]_2$ and L-proline (Scheme 6. eq. 5). By testing this complex with the model reaction, it gave the target product **CI** in 75% yield. In comparison, the use of $[\text{Cp}^*\text{IrCl}_2]_2$ catalyst without L-proline only gave a 48% product yield (Table 1, entry 18). Further, $\text{Cp}^*\text{Ir}(L\text{-Pro})\text{Cl}$ in combination with additional L-proline was unable to improve the product yields (Scheme 6, eq. 6). These experiments clearly indicate that $\text{Cp}^*\text{Ir}(L\text{-Pro})\text{Cl}$ is the reaction catalyst, and L-proline serves as an effective ligand.

2. The exploration of the Ir catalyst should be expanded. $[\text{IrCl}(\text{COD})]_2$ gives low conversion in the absence of *L*-proline, but how does it compare with it? Also, how effective is Cp^*IrCl_2 without any additive? These reactions should be added to Table 1.

Response: Thank you very much for the constructive comments. We have supplemented the relevant experiments. By performing the model reaction in *t*-AmOH without *L*-proline, the Ir(III) complex $[\text{Cp}^*\text{IrCl}_2]_2$ gave the product **C1** in 22% yield (Table 1 entry 1), whereas the Ir(I) complex $[\text{IrCl}(\text{cod})]_2$ produced a very poor yield (Table entry 2, <5%). Thus the Ir(III) catalyst $[\text{Cp}^*\text{IrCl}_2]_2$ was applied for further optimization of reaction conditions. By utilizing the optimal mixed solution, the test of iridium species showed that iridium(I) complexes (e.g., $[\text{IrCl}(\text{cod})]_2$ and $[\text{Ir}(\text{OMe})(1,5\text{-cod})]_2$) are not suitable for the reaction, whereas iridium(III) complexes (Such as $[\text{Cp}^*\text{IrCl}_2]_2$ and $\text{IrCl}_3 \cdot 3\text{H}_2\text{O}$) are applicable for the transformation. In comparison, $[\text{Cp}^*\text{IrCl}_2]_2$ showed better activity (see addition Table 1, entry 19).

3. The work is well done, but the results obtained are not a significant improvement on related reactions published. There are some areas where the language and grammar need improvement, so I recommend some careful revision of the English usage prior to submission in a more specialized journal on organic synthesis or catalysis.

Response: Thank you very much for the comments. Here, we believe that our work is totally different from the previous contributions, and brings sufficient improvements: (1) Our developed catalysis does not need the pre-preparation of specific coupling agents (e.g., quinoline *N*-oxides, organozinc and Grignard agents) and the use of excess less-environmentally benign oxidants and elevated temperatures, and the *N*-heteroarenes is directly employed as the coupling agents. (2) Our

developed chemistry showed far broader substrate scope. In addition to quinoline derivatives, other N-heteroarenes such as quinoxaline, quinazoline, 1,5-naphthyridine, 1,8-naphthyridine, imidazo[1,2-*a*]pyrazine, 7,8-benzoquinoline, phenanthridine, thieno[3,2-*b*]pyridine, and even more challenging pyrimidine, pyrazine, and pyridine derivatives are all compatible with our transformation. Moreover, except for phenyl and furanyl boronic acids, a vast range of heteroaryl boronic acids such as pyridyl, thiophenyl, and other N-heteroaryl ones are all applicable coupling partners. More importantly, our synthetic protocol offers a new platform for direct preparation of N-bidentate ligands that are difficult to prepare with the existed α -C-H arylation approaches. (3) Our work demonstrates a new water-promoted hydrogen evolution coupling mechanism, which has offered the potential for further functionalization of other unsaturated hydrocarbons and design of new catalytic transformations.

Reviewer #2 (Remarks to the Author):

This is a very nice chemistry to carry out the oxidative coupling between heterocycles and arylboronic acids via Ir catalysis. Especially a new pathway was observed by evolving the dihydrogen. The chemistry showed its potential application. Several points must be addressed before publications.

Response: we appreciate the reviewer's positive comments.

1. The substrate scope is relatively narrow although the potential applications have been demonstrated. How about other heterocycles, in particular pyridine derivatives?

Response: In addition to quinoline derivatives, other N-heteroarenes such as quinoxaline, quinazoline, 1,5-naphthyridine, 1,8-naphthyridine, imidazo[1,2-*a*]pyrazine, 7,8-benzoquinoline, phenanthridine, thieno[3,2-*b*]pyridine, and even more challenging pyrimidine and pyrazine all worked well to afford the desired products. We have successfully achieved the α -arylation of pyridine derivatives, albeit the yields are somewhat low (see products **C56-C58** in Scheme 4 in the manuscript.).

2. In the catalytic cycle, H⁺ seems play a role as an oxidant to make the oxidation take place to generate H₂. Please give the clear demonstration in the catalytic cycle.

Response: Thank you for the suggestion. We have made the changes in the catalytic cycle (see Scheme 7 in the manuscript).

3. Final, the reference of the first example to demonstrate the oxidative addition between heterocycles and arylboronic acids might be better cited in the text (Angew. Chem. Int. Ed. 2008, 47, 1473-1476).

Response: Thank you for the suggestion. We have cited this reference as ref 32 in the revised manuscript.

Reviewer #3 (Remarks to the Author):

The manuscript by Zhang and co-workers describes a simple method for the alpha-arylation of nitrogen heterocycles. The compounds available by this method are very interesting from a pharmaceutical point of view and some of them are not readily available by alternative methodologies. Furthermore, the manuscript is well written. I recommend its acceptance with some corrections and clarifications.

Response: Thank you very much for the positive comments.

1. In Scheme 1, the use of a reducing agent should be added to the arrows.

Response: Thank you for the suggestion. We have done so in the revised manuscript (Scheme 2).

2. Have the authors tried the use of pyridine derivatives as substrates? This would lend a higher generality to their method. They have showcased the use of other single-ring heterocycles such as pyrimidines and pyrazines.

Response: Thank you very much for the nice suggestion. We have achieved the application of pyridine derivatives, although the yields are moderate, please see the products **C56-C58** in Scheme 4 in the manuscript.

3. Line 163. Please explain the KIE acronym.

Response: We have explained the acronym of KIE in the revised manuscript.

Reviewers' Comments:

Reviewer #1:

Remarks to the Author:

I am satisfied with the revisions. The paper is suitable for publication.

Reviewer #2:

Remarks to the Author:

After the revision, the quality of the manuscript was high promoted. I would suggest the acceptance of this article as it.

Reviewer #3:

Remarks to the Author:

In this revised version, the authors have nicely implemented the suggestions from the referees. In my opinion, the manuscript is now suitable for publication.